# Telesimulation for the Training of Medical Students in Neonatal Resuscitation

**DOI:** 10.3390/children10091502

**Published:** 2023-09-04

**Authors:** Lukas P. Mileder, Michael Bereiter, Bernhard Schwaberger, Thomas Wegscheider

**Affiliations:** 1Clinical Skills Center, Medical University of Graz, 8010 Graz, Austria; 2Division of Neonatology, Department of Pediatrics and Adolescent Medicine, Medical University of Graz, Auenbruggerplatz 34/2, 8036 Graz, Austria

**Keywords:** neonate, resuscitation, mask ventilation, education, simulation training, telesimulation, remote simulation

## Abstract

**Background:** Telesimulation may be an alternative to face-to-face simulation-based training. Therefore, we investigated the effect of a single telesimulation training in inexperienced providers. **Methods:** First-year medical students were recruited for this prospective observational study. Participants received a low-fidelity mannequin and medical equipment for training purposes. The one-hour telesimulation session was delivered by an experienced trainer and broadcast via a video conference tool, covering all elements of the neonatal resuscitation algorithm. After the telesimulation training, each student underwent a standardized simulated scenario at our Clinical Skills Center. Performance was video-recorded and evaluated by a single neonatologist, using a composite score (maximum: 10 points). Pre- and post-training knowledge was assessed using a 20-question questionnaire. **Results:** Seven telesimulation sessions were held, with a total of 25 students participating. The median performance score was 6 (5–8). The median time until the first effective ventilation breath was 30.0 s (24.5–41.0) and the median number of effective ventilation breaths out of the first five ventilation attempts was 5 (4–5). Neonatal resuscitation knowledge scores increased significantly. **Conclusions:** Following a one-hour telesimulation session, students were able to perform most of the initial steps of the neonatal resuscitation algorithm effectively while demonstrating notable mask ventilation skills.

## 1. Introduction

The COVID-19 pandemic has had a significant and enduring impact on medical education, as universities and educators were suddenly forced to develop and implement novel methods to impart knowledge and to ensure essential hands-on training while being limited by social distancing [1]. Simulation educators often switched their face-to-face programs to virtual simulation training, distance simulation or telesimulation in order to sustain their curricula [2], although there were significant differences in preparedness and successful implementation of distance simulation or remotely facilitated training between geographic regions [3].

Telesimulation uses telecommunication resources and simulation equipment “to provide education, training, and/or assessment to learners at an off-site location” [4]. Its use has increased significantly since 2020, but published studies and program reports have so far mainly focused their evaluations on learners’ reactions instead of higher levels of learning outcomes [5].

In the context of neonatal medicine, only a few applications of telesimulation have been reported so far. A randomized controlled trial comparing neonatal resuscitation education in the classroom with teleinstruction found similar increases in post-training knowledge and skills scores in participating nurses [6]. McCaw et al. [7] reported similar results in skills decay when comparing traditional in-person refresher training with remote instruction after initial Helping Babies Breathe training. In our own pilot study of one-hour telesimulation sessions for the training of neonatal resuscitation, we found telesimulation to be feasible and to be associated with significant improvements in guideline knowledge among student near-peer-teachers and experienced neonatal nurses [8]. As age and experience may influence perception and the impact of simulation-based learning [9], we investigated in this follow-up study the effect of a single telesimulation training on the performance of simulated neonatal resuscitation in inexperienced providers.

## 2. Materials and Methods

For this purpose, we performed a prospective observational study. Participants were first-year medical students who were invited by e-mail to participate in this study after having completed a training in basic clinical and procedural skills [10] at the Clinical Skills Center, Medical University of Graz, Austria, including a two-hour training on airway management and bag-valve-mask ventilation in adults. Students were informed by e-mail and verbally (BS, LPM) about the specific study aims and how they would be evaluated before and after training participation. Study participation was voluntary without financial compensation, and written informed consent was obtained from all of the participants.

As no previous study has investigated the impact of telesimulation training in inexperienced neonatal resuscitation providers, we were not able to perform a sample size calculation. Therefore, we aimed for an arbitrary number of 30 study participants.

### 2.1. Telesimulation Training

All training interventions were delivered remotely for groups of 2–4 students. For individual training, participants were presented with a neonatal resuscitation training kit: a low-fidelity mannequin (Newborn Anne™ or Baby Anne™, Laerdal Medical, Stavanger, Norway, depending on availability), towel, plastic cap, stethoscope, suction catheter, and a neonatal self-inflating ventilation bag with appropriately sized face mask (both Laerdal Medical, Stavanger, Norway). Training took place in a seminar room at our university’s campus. There, students were separated from each other and participated in the telesimulation training via an individual personal computer with camera and earphones.

Each telesimulation session lasted for 60 minutes and was broadcast via Cisco Webex (Webex by Cisco, San José, CA, USA) from the neonatal resuscitation suite at our Clinical Skills Center. Every training was delivered by an experienced trainer (LPM) as an algorithm training with open discussion. Telesimulation covered all elements of the recent neonatal resuscitation algorithm [11], containing deliberate practice of drying, tactile stimulation and opening the airway, bag-valve-mask ventilation, and chest compressions. Participants demonstrated skills in consecutive order, and they received individual feedback on task performance. All participants of each telesimulation session were able to watch their peers’ performance and also to hear the trainer’s feedback, which further added to the learning effect though passive observation.

### 2.2. Assessment

Within one hour after the telesimulation training and after having answered the post-training questionnaire, each student individually underwent a standardized simulated scenario involving a non-vigorous term neonate after birth, who required mask ventilation for 60 s. For assessment purposes, each student underwent the scenario alone. The scenario was conducted at the neonatal resuscitation suite of our Clinical Skills Center using a Newborn Anne™ (Laerdal Medical, Stavanger, Norway). Every student’s performance was video-recorded, and the scenario videos were evaluated by an uninvolved neonatologist (BS), who rated the performance using a composite score (maximum: 10 points) based on the first measures recommended for non-vigorous neonates after birth (drying/wrapping and stimulation, keeping warm, heart rate assessment: auscultation/ECG, opening the airway, giving five inflation breaths) [11]. Tasks were scored with zero points if omitted or incorrectly performed, with one point if partially performed or performed in the wrong sequence, and with two points if performed effectively and in the correct sequence.

Furthermore, we evaluated (i) the time from arrival at the resuscitation table until the first (effective) ventilation breath was delivered, (ii) the number of effective ventilation breaths out of the first five ventilation attempts, (iii) the number of “prolonged” ventilation breaths (inspiratory time 2–3 s) out of the first five ventilation attempts, and (iv) the number of ventilation breaths until five were delivered effectively.

To measure the impact of telesimulation training on students’ knowledge of current neonatal resuscitation guidelines [11], we used a previously composed 20-question single-choice paper-and-pencil questionnaire [8], which was distributed among students before and after the training.

### 2.3. Statistical Analysis

We used SPSS Statistics 28 (IBM Corporation, Armonk, NY, USA) to analyse the data. Frequencies are displayed using absolute and relative numbers. Continuous variables are given as median (interquartile ranges [IQR]). We used Wilcoxon-rank-sum test to compare pre- and post-training knowledge scores. A *p*-value of <0.05 was considered statistically significant.

## 3. Results

Ultimately, 25 medical students (median age: 22 [IQR 20–24] years; median semesters of studies: 2 [IQR 1–2]; m:f = 16:9) participated in the study, none of whom had actual experience in postnatal resuscitation. A total of seven telesimulation sessions were held successfully between February 2022 and April 2023.

The median performance score was six points (5–8). Results of the video assessment for the five individual tasks are summarized in Table 1.

The median time from arrival at the resuscitation table until the first ventilation breath was 26 s (22–39). A total of 23 (92%) of the 25 students initiated positive pressure ventilation within 60 s (Figure 1), and the median time from arrival at the resuscitation table until the first effective ventilation breath was 30 s (24.5–41.0). The median number of effective ventilation breaths out of the first five ventilation attempts was 5 (4–5; Figure 2), but the number of “prolonged” ventilation breaths out of the first five ventilation attempts was low (median 0 [0–5]). The median number of ventilation breaths until five were delivered effectively was 5 (5–6).

After the telesimulation training, neonatal resuscitation knowledge scores increased significantly from a median of 12.0 (10.5–14.0) before to 19 correct answers (18–20) after the educational intervention (*p* < 0.001).

## 4. Discussion

Our study showed that participation in a one-hour telesimulation training improved resuscitation competence in inexperienced providers. Following the educational intervention, medical students did not only have significantly improved knowledge of current neonatal resuscitation guidelines [11], but they were also able to adequately execute the main elements of the neonatal resuscitation algorithm in a simulated setting. This supports results from previous investigations, showing improved procedural competence following telesimulation training, e.g., in pediatric resuscitation [12], gaining intraosseous vascular access [13], and chest tube insertion [14].

Performance ratings from video review showed a median performance score of six out of a maximum of ten points. The main deficit was found in temperature control, which is an essential therapeutic intervention to improve neonatal health. Postnatal hypothermia has a dramatic impact on neonatal outcome, as the mortality risk may increase by 80% for every degree of Celsius decrease, as shown for neonates in Nepal [15]. Although the importance of maintaining normothermia after birth was emphasized during telesimulation training and a plastic cap and warm and dry towels were provided for the simulated scenario, students primarily focused on other aspects of the neonatal resuscitation algorithm, which shows that more attention to this topic is warranted during future trainings.

The finding that students were able to deliver the first effective ventilation breath within 30 s after arrival at the resuscitation table is important, as this compares well to the European Resuscitation Council’s recommendation to “start positive pressure ventilation as soon as possible–ideally within 60 s of birth” [11]. This underscores the finding that even short practical training interventions are associated with improved mask ventilation performance [16,17]. Most of the students were able to achieve effective chest rise during the first five ventilation attempts, again emphasizing the impact of this brief telesimulation training. Significant face mask leak, airway obstruction, and challenges to appropriately estimate applied tidal volumes often render non-invasive ventilation difficult for healthcare professionals charged with postnatal care [18], and telesimulation may be one option to facilitate regular refresher training in order to mitigate the decay of skills [7].

While students showed high levels of competence in bag-valve-mask ventilation, delivered ventilation breaths were generally short ones and not, as recommended by the European Resuscitation Council, delivered with the inflation pressure maintained “for up to 2–3 s” [11]. This finding is rather surprising, as the “prolonged” initial ventilation breaths were specifically explained and taught during telesimulation training. One explanation could be that students were probably more familiar with pediatric and adult resuscitation algorithms, both of which recommend short inspiratory times of approximately one second even for the initial ventilation breaths [19,20]. Furthermore, maintaining the inflation pressure for 2–3 s is challenging in the presence of relevant face mask leak, which is more frequent among inexperienced providers [21]. Finally, it is important to note that the delivery of initial ventilation breaths with inspiratory times of 2–3 s is not unequivocally agreed upon––the American Heart Association recommends “to initiate PPV [positive pressure ventilation] at a rate of 40 to 60/min to newly born infants who have ineffective breathing” and that “the inspiratory time while delivering PPV should be 1 s or less” [22].

In our opinion, telesimulation training shall not replace traditional face-to-face simulation-based training such as the Neonatal Resuscitation Program [23] but should rather augment it, serving as a link between theoretical instruction, technical skills practice, and scenario-based algorithm training. Informal talks with the students following their study participation clearly showed that participants valued the educational experience but would not want to have telesimulation replace face-to-face instruction. This has also been concluded by Löllgen et al. [2], whose questionnaire-based study among distance simulation participants illustrated high overall satisfaction, despite limitations such as technical issues, time delays, and the (required) spatial separation. Another challenge when delivering telesimulation training can be the quality of instructor feedback [8], which requires trainers to develop “interactive skills” [24].

Studies directly comparing telesimulation to face-to-face simulation-based training are still scarce. In their review including 29 articles, Yasser et al. [5] identified only two studies with a pre-post-test design and two randomized controlled trials. McCoy et al. [25] found similar mean performance scores in medical students following either telesimulation or in-person simulation-based training in two scenarios involving adults suffering from cardiac arrest and anaphylaxis. Accordingly, Jewer et al. [14] reported improved technical skills performance and similar satisfaction among participants when comparing telesimulation training using a mobile unit to face-to-face simulation-based training. Based on these encouraging findings, we are optimistic that the improved technical skills level in our study would compare well to a traditionally trained group of students.

In our opinion, future research on telesimulation should address five main aspects: (i) the best practices of telesimulation training related to technology and didactics, (ii) direct comparison with face-to-face simulation-based training, both as an initial and refresher training modality, (iii) skills maintenance after telesimulation training compared with face-to-face simulation-based training, (iv) telesimulation for the training of non-technical skills and team behavior, and (v) the transfer of skills to actual patient care after telesimulation training.

### Limitations

Our study’s main methodological limitations are the rather small sample size and the lack of a control group.

Secondly, to allow for an unbiased performance assessment, we decided that students should undergo the post-telesimulation assessment on their own. We are fully aware that effective neonatal resuscitation requires a team approach [26] and, therefore, can only speculate how neonatal resuscitation performance would have been impacted by a two-provider setting.

Thirdly, video recordings of students’ post-training performance were evaluated by an experienced neonatologist and simulation trainer, but having two experts independently review the videos may have further improved the reliability of our findings [27].

Fourthly, we cannot completely rule out a potential impact of the bag-valve-mask ventilation training that all students underwent as part of their regular curriculum before participation in this study [10]. However, the aforementioned ventilation training addressed only adults and we believe its effect to be small, as neonatal airway anatomy, respiratory dynamics, and their clinical implications differ significantly [28].

Fifthly, the finding of improved students’ ventilation performance must be interpreted with caution, because manual ventilations in mannequins may not equal clinical competence, as mannequins are rigid, lack secretions, and thus allow for a more effective face mask positioning [29]. Nonetheless, one should bear in mind that skills acquired in the simulated environment may be successfully transferred to patient care [30,31,32].

Sixthly, we only assessed the impact of telesimulation immediately after the training. Based on this study, we are not able to predict the maintenance of skills and their decay over time following a single telesimulation session [33].

Finally, our study focused on inexperienced providers and the initial steps of the neonal resuscitation algorithm. We can only speculate to what extent telesimulation could also be used to train professionals with more experience in more complex (non-technical) skills, such as team communication, leadership, and crisis resource management [34].

## 5. Conclusions

Following a single one-hour telesimulation session, first-year medical students were able to perform most of the initial steps of the neonatal resuscitation algorithm effectively while demonstrating notable bag-valve-mask ventilation skills in the simulated setting. These findings, although derived from a small sample size, suggest that telesimulation can be used to teach and train neonatal resuscitation skills to novice providers.

## Figures and Tables

**Figure 1 children-10-01502-f001:**
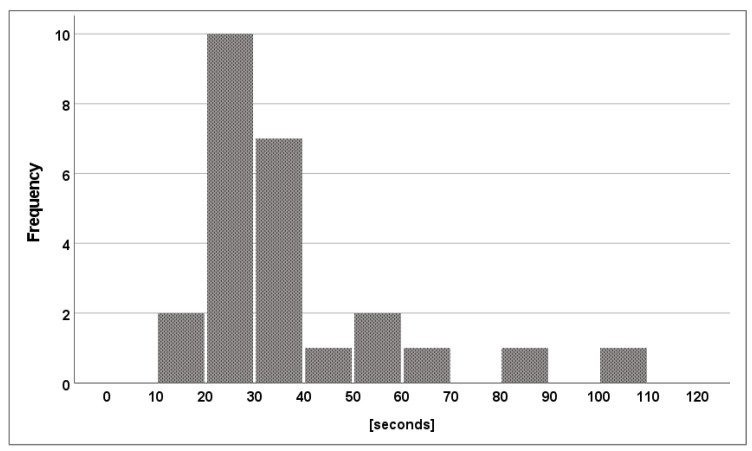
Time from arrival at the resuscitation table until the first effective ventilation breath.

**Figure 2 children-10-01502-f002:**
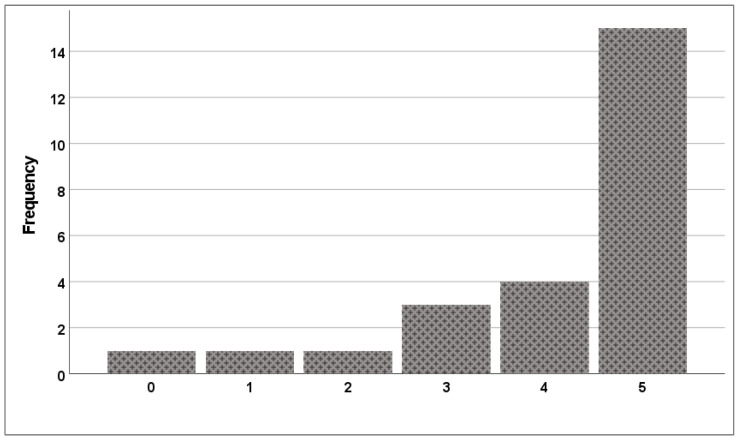
Number of effective ventilation breaths out of the first five ventilation attempts.

**Table 1 children-10-01502-t001:** Results of students’ performance of initial tasks based on video review (0 points = task omitted or incorrectly performed; 1 point = task partially performed or performed in wrong sequence; 2 points = task performed effectively and in correct sequence).

Task	Median Points	IQR
Drying/wrapping and stimulation	2.0	0.5–2.0
Keeping warm *(*check preheated radiant warmer, remove wet linen, cover neonate’s head and body*)*	0	0.0–1.5
Heart rate assessment: auscultation and/or ECG	1	0–2
Opening the airway	2	1–2
Giving five inflation breaths	1	1–2

## Data Availability

The data presented in this study are available on request from the corresponding author.

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
