# Peer review of "Telesimulation for the Training of Medical Students in Neonatal Resuscitation"

_children, 2023, doi:10.3390/children10091502_

Round 1

Reviewer 1 Report

This is a study utilizing telesimulation to train first year medical students. The sample size is very small (25) at one center.  The authors could work to more clearly explain their methods that it was done in different phases and all of the training was done remotely. Was the 1 hour session done 1:1? Or was it a group setting? Was the student doing a resuscitation alone in the simulation case? Is this supposed to replace NRP teaching? Is there any comparison available for teaching in an in-person setting?

In addition, given the small sample size, the conclusion is a little too strong. There should be suggestions about further work or adaptation into different settings/scenarios. The discussion should include work that exists on efficacy of in-person compared to virtual training, if it is possible to find relevant research.

Reviewer 2 Report

The authors have presented their brief report on telesimulation for the training of medical students in neonatal resuscitation. As noted in the limitations, this study is limited by small sample size. However by adding some of the followings listed below, the significance of this study in resuscitation  education may be enhanced. 

1. In the introduction, the authors should elaborate on how this study is different to the author's previous paper. The purpose is not different from the "Telesimulation as a modality for neonatal resuscitation training" paper other than substituting the study population. It would be interesting to add the author's reasoning on how and why this study was planned as an extension.

2. Analysis of descriptive feedback from the medical students on telesimulation would be helpful. It would be interesting to know how this methodology affected their learning compared to other educational modalities such as lectures, hands on simulation and etc . 

3. It would be interesting to see how much knowledge and technical skills were retained after a set time period. Telesimulation and evaluation through simulated neonatal resuscitation seemed to have taken place immediately.  Retainment of knowledge and skills may differ in a single day telesimulation.

4. Despite the small number of students, it would be interesting to analyze factors associated with higher knowledge test scores and performance scores. This may provide insight into finding the characteristics of students best fit for telesimulation. 

Reviewer 3 Report

The authors present an educational study that evaluates medical student knowledge and performance of the initial steps of neonatal resuscitation in a simulation scenario after participation in a telesimulation training session.  The COVID pandemic highlighted the need to develop effective methods of distance learning, and this study helps contribute to the growing body of work on the use of telesimulation.  While studies have demonstrated the potential use of telesimulation, its effectiveness with a novice group of learners requires further investigation. 

I have several questions and comments related to the study design, results and discussion/conclusion.

Methods

1. I suggest providing additional information on how the study was presented to the students, as it would help the reader interpret the study results.  Did they know the specific study aims and how they would be evaluated before participation?

2. From my reading, even though the students simultaneously participated in the telesimulation training session and observed the instructor in groups of 2-4 learners, they had individual headsets and computers.  If this is the case, when the instructor provided feedback on student performance of skills, were the other students able to observe each other performing the skills and learn by observing as well?  This may be a consideration for future educational sessions.

3. I appreciated details about data analysis.  It would also be helpful to see a sample size calculation, especially since the study had a small sample size.  If there are no prior studies that would aid in a sample size calculation, then providing information about an effect size would have been helpful.

4. To increase the robustness of the performance evaluation, it may have been helpful for the students to participate in another scenario that requires performance of the same skills but may vary in the complexity of prenatal history of infant presentation.  It may have also been interesting to see student performance in pairs, especially since neonatal resuscitation emphasizes a team approach to assessment and infant stabilization.

5. There was only one investigator who rated all the videos.  To increase robustness, consider for future studies a second rater to assess inter-rater reliability.

Results

1. The results were clear.  It is interesting that the students had better performance of bag-mask ventilation than of the steps to provide warmth.  As part of demographic information, it may be useful to know whether they have participated in any other life support courses where they received education about providing ventilation.

2. For the evaluation of thermoregulation, were students required to perform all four steps (preheating the radiant warmer, removing wet linen, etc.) to receive the full two points?  That may be another reason why they were not able receive any points.

3. It is not surprising that the students improved on their post-telesimulation knowledge assessment since it was provided within an hour of the training session.  Out of curiosity, were there questions that most students answered incorrectly before training?

Discussion/Conclusion

1. Lines 189-193: Based on your study design, I suggest clarifying in your discussion and conclusion sections that telesimulation may be utilized to teach novice learners the initial steps of neonatal resuscitation.  It is still unclear whether it would be as effective in teaching more advanced skills, especially those that require the coordination of a full team and those that involve more complex kinesthetic skills.

Other

1. Line 41: I am not sure I completely understand the use of the word “non-presential,” which I think means not at the present time.  However, given the context of the sentence, do you mean to say not in the same location?

Round 2

Reviewer 1 Report

The paper is improved from the previous iteration. The limitations section could be written to be slightly more fluid but otherwise points out a more appropriate list of limitations.  In addition, the study could be improved by also adding 'next steps' for potential research.

Reviewer 3 Report

In this revised manuscript, the authors provided additional clarifying details that I feel helped to strengthen the methods, discussion, and conclusion sections.  I found one typo for the word "program" in line 212.
